# Characterization of Fe_3_O_4_ Nanoparticles for Applications in Catalytic Activity in the Adsorption/Degradation of Methylene Blue and Esterification

**DOI:** 10.3390/molecules27248976

**Published:** 2022-12-16

**Authors:** Juan Sebastian Trujillo Hernandez, Alberto Aragón-Muriel, Willinton Corrales Quintero, Juan Camilo Castro Velásquez, Natalia Andrea Salazar-Camacho, German Antonio Pérez Alcázar, Jesús Anselmo Tabares

**Affiliations:** 1Grupo de Metalurgia Física y Teoría de Transiciones de Fase, Departamento de Física, Universidad del Valle, Cali 25360, Colombia; 2Centro de Excelencia de Nuevos Materiales (CENM), Universidad del Valle, Cali 25360, Colombia; 3Laboratorio de Investigación en Catálisis y Procesos (LICAP), Departamento de Química, Universidad del Valle, Cali 25360, Colombia; 4Facultad de Ciencias Naturales y Matemáticas, Universidad de Ibagué, Ibagué 730002, Colombia

**Keywords:** Fe_3_O_4_ nanoparticles, catalytic activity, Mössbauer spectroscopy, methylene blue, esterification

## Abstract

The aim of this study is to evaluate the applicability of the catalytic activity (CA) of the Fe_3_O_4_ magnetic system in the adsorption/degradation of methylene blue and esterification. The thermal decomposition method allowed the preparation of Fe_3_O_4_ nanoparticles. The crystallites of the Fe_3_O_4_ structural phase present an acicular form confirmed by X-ray diffraction. Transmission electron microscopy results identified the acicular shape and agglomeration of the nanoparticles. Mössbauer spectroscopy showed that the spectrum is composed of five components at room temperature, a hyperfine magnetic field distribution (HMFD), two sextets, a doublet, and a singlet. The presence of the HMFD means that a particle size distribution is present. Fluorescence spectroscopy studied the CA of the nanoparticles with methylene blue and found adsorption/degradation properties of the dye. The catalytic activity of the nanoparticles was evaluated in the esterification reaction by comparing the results in the presence and absence of catalyst for the reaction with isobutanol and octanol, where it is observed that the selectivity for the products MIBP and MNOP is favored in the first three hours of reaction.

## 1. Introduction

Iron oxides are widely distributed in nature and can be easily synthesized in the laboratory [1,2,3,4,5]. There are 16 iron oxides, one of the most common being magnetite (Fe_3_O_4_), and they can be present in their two aggregation states, as Fe^+2^ and Fe^+3^ forming oxides [6,7,8,9]. Most iron oxides are crystalline. Physical and chemical properties depend on variables such as the degree of structural order and size of the crystal [7,8,9,10]. Magnetite is an iron oxide noted for its magnetic properties. Magnetite nanoparticles exhibit a large surface area that allows them to be functionalized to make biocompatible products (porous surface) [4,5,6,7,8,9,10,11]. This characteristic makes them especially suitable for a wide range of applications in diverse areas such as biotechnology, environmental, medical, pharmaceutical, catalysts, pigments, gas sensors, optical and electromagnetic devices, decontaminants, and wastewater treatment for dye removal [7,8,9,10,11,12,13]. In many chemical synthesis processes, the use of catalysts is necessary to carry out a specific reaction, so both conversion and selectivity are important criteria to evaluate the activity of a catalyst. Among the chemical reactions that use catalysts are esterification reactions where an ester and water are obtained from a carboxylic acid and an alcohol. The use of metal oxide nanoparticles as catalysts in esterification reactions has been evidenced [11,12,13,14,15,16,17,18], which has generated great interest in developing new alternative materials that can be used as more active and low-cost catalysts. In applications for the removal of contaminants containing iron oxide (Fe_3_O_4_) nanoparticles, Mössbauer studies have shown that magnetite (Fe_3_O_4_) spectra taken at room temperature show magnetic (ferromagnetic) behavior represented by two sextets, characteristic of the Fe^+2^ (octahedral sites) and Fe^+3^ (tetrahedral sites) states, for larger particle sizes (greater than 100 nm) [19,20,21]. However, the spectrum shows a non-magnetic (paramagnetic–superparamagnetic) behavior, represented by a doublet or singlet for particle sizes below a critical size (smaller than 20 nm) [21].

In this work, magnetite nanoparticles were produced from the thermal decomposition method, obtaining a particle size distribution where the magnetic measurements showed a competition between ferromagnetic and superparamagnetic behavior. By means of fluorescence spectroscopy, the catalytic activity in the adsorption/degradation of methylene blue was evidenced. The catalytic activity for the esterification of phthalic anhydride with isobutanol and octanol was also investigated; the nanostructured Fe_3_O_4_ powders could effectively catalyze the esterification.

## 2. Results

### 2.1. Structural Results

Refinement of the XRD results (see Figure 1) shows a spinel structure according to the lattice parameters of magnetite a = b = c = 8.379 Å [10,17,22], indicating an acicular shape of the crystals with parallel and perpendicular crystallite sizes of 79.5 ± 1 Å and 137.2 ± 2 Å, respectively [17]. According to the results obtained by TEM (see Figure 2), a tendency toward agglomeration is present in Fe_3_O_4_ nanoparticles with grain sizes smaller than 50 nm.

### 2.2. Mössbauer Results

Figure 3 shows the fit of the Mössbauer spectrum of Fe_3_O_4_. The fit employed in the spectrum (singlet, doublet, sextet, and a hyperfine magnetic field distribution—HMFD), evidences a particle size distribution given by a HMFD with isomer shift (IS) of −0.22 ± 0.05 mm/s, mean hyperfine field (B_hf_) of 39.1 ± 0.2 T, and linewidth (GA) of 0.30 ± 0.05 mm/s [17,19,20]; the two sextets are associated with iron ions (Fe^+2^ and Fe^+3^), located at different crystallographic positions within the spinel structure of Fe_3_O_4_ with IS of 0.41 ± 0.05 and −0.23 ± 0.05 mm/s, B_hf_ of 39.2 ± 0.2 and 47.4 ± 0.2 T. These ionic states at room temperature are not discernible due to their fast electron hopping [17,19,20]. The presence of the two sextets with Mössbauer parameters similar to those reported for the bulk magnetite is attributed to the larger particles. The central part of the spectrum, which contributes the majority of spectral area, is due to the smaller particles, some of which present low hyperfine fields and are near the transition from the superparamagnetic to ferrimagnetic state, and also to those smallest nanoparticles that behave as superparamagnetic and appear as a doublet or singlet [17,21].

### 2.3. FT-IR Spectra

The results obtained by FT-IR are shown in Figure 4. In the spectrum, two strong bands are observed at 669 and 595 cm^−1^, due to the stretching of the Fe-O (metal–oxygen) bonds by the vibrations in the crystal lattice of Fe_3_O_4_ [7,8,9,22,23,24,25,26], characteristically pronounced for ferrites. The octahedral and tetrahedral sites of the oxide are related to these vibration bands. Additionally, hydroxyl groups attached to hydrogen bonds are present in other bands within the spectrum (commonly of the H_2_O molecule) [8,9].

### 2.4. Catalysis Studies

The adsorption/degradation rate of MB was obtained by the catalytic activity of the sample in the presence of Fe_3_O_4_ nanoparticles. To carry out this study over time, it was necessary to choose the optimal dose of nanoparticles, which was achieved by varying the Fe_3_O_4_/MB ratio in different experiments by UV-Vis spectrophotometry for 1 h of interaction. Figure 5 shows that from 2 mg Fe_3_O_4_/10 mL MB solution (20 mg/L), there is a significant change in the decrease in absorption corresponding to the presence of the dye; therefore, the following studies were carried out using this dose.

Figure 6 shows at different adsorption times of Fe_3_O_4_ the degradation of MB at a dose of 2 mg Fe_3_O_4_/10 mL MB solution. The emission spectra indicated that the concentration of MB decreases with decreasing emission band at 680 nm, showing unique properties in the adsorption/degradation of this cationic dye with the absorption time of the nanoparticles. Photocatalytic studies with UV irradiation, can increase the catalytic activity of the sample [9,10,25]. Effectively, the nanoparticles showed dye adsorption properties at the time of carrying out the study by fluorescence spectroscopy, since at the beginning a decrease in the concentration of methylene blue was observed without the formation of byproducts (confirmed by mass spectrometry). However, after several minutes of interaction of the nanoparticles with the dye, several methylene blue degradation products were identified in the solution, such as benzenesulfonic acid and thionin, molecules that have been found in other studies of the degradation of methylene blue using mass spectrometry [26], so it can be suggested that the nanoparticles presented both adsorption and catalytic degradation properties.

To understand the adsorption/degradation rate of the dye using synthesized nanoparticles, it was necessary to apply a kinetic model that involves the two simultaneous processes, so it is not recommended to use first- and second- order models or Langmuir–Hinshelwood models for correlation [27,28]. On this occasion, a model similar to the one reported by Giovannetti et al. was used for the adsorption and degradation processes separately [28]. The transformation of MB into adsorbed MB (MB_Fe3O4_) and into photodegraded products (MB_PH_) could be treated in terms of two consecutive processes. The first is represented by the decrease in the concentration of MB in the solution due to the adsorption on the surface of Fe_3_O_4_ nanoparticles. It is shown that the absorption of MB occurs with a process described by the first order kinetic constant *k_1_* expressed by Equation (1).
(1)$$ln\left[\frac{\left(q_{e}-q_{t}\right)}{q_{t}}\right]=-k_{1}t$$
where *q_t_* is the amount of dye adsorbed at time *t* and *q_e_* is the equilibrium concentration [29]. The second process is represented by the photodegradation of MB, a process that occurs with a speed described by the first order kinetic constant *k*_2_ expressed by Equation (2) [29,30].
(2)$$ln\left[\frac{C}{C_{0}}\right]=-k_{2}t$$
where *C*_0_ is the initial concentration of MB and *C* the concentration of dye at time *t*.
$$MB+Fe_{3}O_{4}\to MB_{Fe_{3}O_{4}}\;\;\;\;\;\;\;\;\;\;\;\;\;\;\;\;\;\;\;\;\;\;\;\;\;\;k_{1}$$
$$MB_{Fe_{3}O_{4}}+h\nu \to MB_{PH}\;\;\;\;\;\;\;\;\;\;\;\;\;\;\;\;\;\;\;\;\;\;\;\;\;\;\;\;k_{2}$$

The rate at which MB decreases and the rate for MB_Fe3O4_ and MB_PH_ formation can be expressed as follows:(3)$$-\frac{d\left[MB\right]}{dt}=k_{1}\left[MB\right]$$
(4)$$\frac{d\left[MB_{Fe_{3}O_{4}}\right]}{dt}=k_{1}\left[MB\right]-k_{2}\left[MB_{Fe_{3}O_{4}}\right]$$
(5)$$\frac{d\left[MB_{PH}\right]}{dt}=k_{2}\left[MB_{Fe_{3}O_{4}}\right]$$

The integration of Equation (3) gives:(6)$${\left[MB\right]}_{t}={\left[MB\right]}_{0}e^{-k_{1}t}$$
where [*MB*] = [*MB*]_0_ at time 0, and [*MB*] = [*MB*]_t_ at time *t*. By substituting Equation (6) into Equation (4), a linear differential equation can be obtained:(7)$$\frac{d\left[MB_{Fe_{3}O_{4}}\right]}{dt}=k_{1}{\left[MB\right]}_{0}e^{-k_{1}t}-k_{2}\left[MB_{Fe_{3}O_{4}}\right]$$
that, after integration, can be written as:(8)$$\left[MB_{Fe_{3}O_{4}}\right]=\frac{k_{1}}{k_{2}-k_{1}}\left(e^{-k_{1}t}-e^{-k_{2}t}\right){\left[MB\right]}_{0}$$
where [*MB*]_0_ = [*MB*] + [*MB_Fe_*_3*O*4_] + [*MB_PH_*], so [*MB_PH_*] = [*MB*]_0_ − [*MB*]*_t_* − [*MB_Fe_*_3*O*4_]*_t_* at time *t*. Substituting [*MB*]*_t_* and [*MB_Fe_*_3*O*4_]*_t_* with Equations (6) and (8), the Equation (9) can be obtained:(9)$$\left[MB_{PH}\right]=\left\{1+\frac{k_{1}e^{-k_{2}t}-k_{2}e^{-k_{1}t}}{k_{2}-k_{1}}\right\}{\left[MB\right]}_{0}$$

Using Equations (6) and (9), it was possible to show that the experimental results present a good correlation with the theoretical values of [*MB*]*_t_* at any time, using concentrations up to 20 mg/L, as shown in Figure 7. The results indicate that as [*MB*]*_t_* decreases towards low concentrations, the concentration of the intermediate *MB_Fe_*_3*O*4_ (calculated from Equation (8)) rises to a maximum and then decreases towards low concentrations, while the concentration of *MB_PH_* (calculated from Equation (9)) increases from the lowest concentration towards [*MB*]_0_. With the above, it is shown that the applied model could explain the experimental results and that, from the calculation of [*MB_Fe_*_3*O*4_] and [*MB_PH_*], the evolution of the adsorption/degradation process could be predicted.

On the other hand, the esterification of phthalic anhydride with isobutanol and octanol for the synthesis of diisobutyl phthalate (DIBP) and di-n-octylphthalate (DNOP), respectively, was investigated due to its important applications, such as plasticity agents, nail polish, explosive materials, and the lacquer industry [23,24,25,26,31]. In the esterification reaction, the chemical reaction shown in Figure 8 occurred, and it was possible to identify the monosubstituted (MIBP, MNOP) and disubstituted (DIBP, DNOP) products.

The catalytic activity of the nanoparticles was evaluated in the esterification reaction by comparing the results in the presence and absence of catalyst for the reaction with isobutanol and octanol, as shown in Figure 9a and Figure 10a, respectively. The selectivity for products MIBP and MNOP is favored in the first three hours of reaction, whereas the concentration of DIBP and DNOP products increases as time progresses. In all cases, a slight increase in the conversion to products is observed, so magnetite nanoparticles exhibit activity in this type of reaction, which could increase at different temperatures or different PA:alcohol ratio or by functionalizing the nanoparticles with other materials.

Finally, in evaluating the effect of catalyst concentration, it can be seen in both Figure 9b and Figure 10b that the best results are obtained for the lowest catalyst:substrate ratios (1:15), so it is clear that the higher the catalyst concentration, the higher the reaction percentage in the esterification reactions studied. It is also important to highlight that in the case of the esterification reaction with isobutanol, not much difference is observed for the results at molar ratios 1:50 and 1:100 (Cat:PA), without even exceeding 50% conversion after 8 h, while for the reaction with octanol, all the concentrations evaluated exceed 50% conversion after 6 h of reaction, so it is suggested that these Fe_3_O_4_ nanoparticles may present better results in the esterification of aliphatic alcohols than longer chain or higher molecular weight alcohols.

## 3. Experimental Section

### Synthesis, Characterization and Catalytic Studies of Fe_3_O_4_ Nanoparticles

Sodium oleate (5.1 mmol, 1851.2 mg) and iron chloride tetrahydrate [FeCl_2_·4H_2_O] (1.4 mmol, 278.3 mg) were stirred in a mixture (280 mL) of water:hexane:ethanol (3:7:4) in a two-necked round bottom flask. The solution was heated for 4 h at 70 °C. The organic layer containing the Fe_3_O_4_–oleate complex at the end of the reaction was washed with distilled water (30 mL) three times. The hexane was evaporated after washing; the resulting Fe_3_O_4_–oleate complex was placed in a Pyrex tube and heated for 4 h at 380 °C at a vacuum pressure of 40 Pa. A black fluid was obtained by cooling the reaction mixture to room temperature. To precipitate the produced Fe_3_O_4_ particles, they were first dispersed in chloroform and dissolved in hexane (20 mL). The particles were isolated by centrifugation and the precipitate appeared upon addition of dispersing agent (chloroform, 0.02 mL). A black solid product was obtained from the precipitate, and this was washed with ethanol and re-dispersed with hexane and finally stored under argon atmosphere. The characterization of the Fe_3_O_4_ sample was first performed by X-ray diffraction (XRD), using Cu-Kα radiation, and the structural parameters were refined by the Rietveld method, using the GSAS program [32]. Secondly, by Mössbauer spectroscopy (MS), the spectra were adjusted using the MOSFIT program [33], using a ^57^Co(RH) source in transmission geometry. Thirdly, by transmission electron microscopy (TEM), the morphology of the Fe_3_O_4_ sample was studied. Fourthly, spectra were obtained by Fourier transform infrared spectroscopy (FTIR), using a NI-COLETT 6700 spectrometer. The electronic absorption spectra were recorded in an UV-visible Evolution 220 Thermo Scientific Spectrophotometer. Finally, using a fluorescence spectrometer (FP-8500, JASCO), the degradation activity of Fe_3_O_4_ was evaluated using 2 mg of magnetite nanoparticles for every 10 mL of a cationic dye (MB) solution (20 mg/L) for the cationic dye (MB), with the emission band at 680 nm (λexc = 633 nm), keeping adsorption experiments constant at 25.0 ± 0.1 °C and 130 rpm, The catalytic esterification reactions were carried out in a two-necked flask; phthalic anhydride (650 mg) and Fe_3_O_4_ nanoparticles (7.5 mg) were added into 2 mL of isobutanol or octanol, and the mixture was heated at 120 and 150 °C, respectively, for several times under stirring. Analysis of the content of the reaction mixtures were made by using a Hewlett Packard 6890 gas chromatograph with a flame ionization detector. The reaction products were identified by mass spectrometry (Shimadzu-GCMS-QP2010) by Electronic Impact ionization at 70 ev.

## 4. Conclusions

In this work, Fe_3_O_4_ nanoparticles were prepared by thermal decomposition. The results obtained by Mössbauer for the Fe_3_O_4_ nanoparticles present the two sextets associated with iron ions (Fe^+2^ and Fe^+3^) and a particle size distribution given by a HMFD at room temperature. The results obtained by UV-Vis spectrophotometry show that there is a significant change in the decrease in absorption of MB at a dose of 20 mg/L. The emission spectra indicated that the concentration of the cationic dye decreases with decreasing emission band at 680 nm, showing unique properties in the adsorption/degradation with the interaction time of the nanoparticles. According to the kinetic model applied to the adsorption/degradation results, it is suggested that the transformation of MB into adsorbed MB (MB_Fe3O4_) and into photodegraded products (MB_PH_) could be treated in terms of two consecutive processes, in which, as time passes, the concentration of MB_PH_ increase while the intermediate product (MB_Fe3O4_) decreases. The catalytic activity of the nanoparticles was evaluated in the esterification reaction by comparing the results in the presence and absence of catalyst for the reaction with isobutanol and octanol. The selectivity for products MIBP and MNOP is favored in the first three hours of the reaction, whereas the concentrations of the DIBP and DNOP products increase over time. Therefore, the magnetite nanoparticles exhibit activity in this type of reaction.

## Figures and Tables

**Figure 1 molecules-27-08976-f001:**
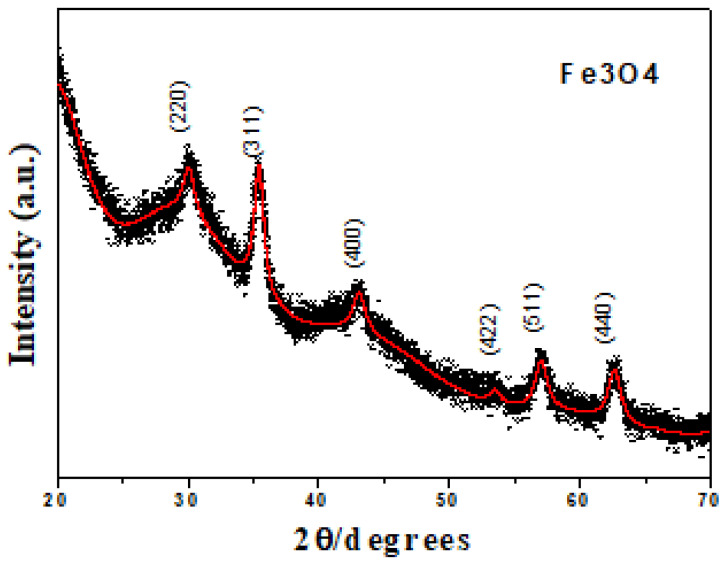
XRD of the Fe_3_O_4_ system at room temperature [17].

**Figure 2 molecules-27-08976-f002:**
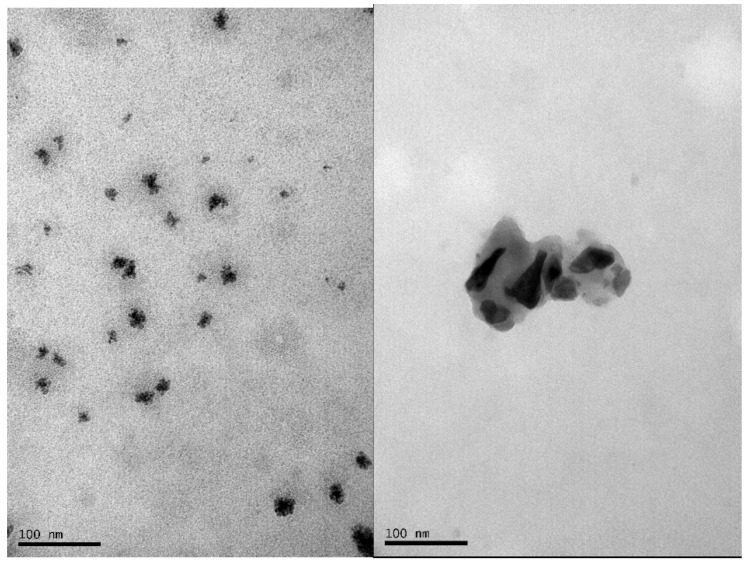
Transmission Electron Microscopy TEM micrograph of Fe_3_O_4_ agglomerated nanoparticles.

**Figure 3 molecules-27-08976-f003:**
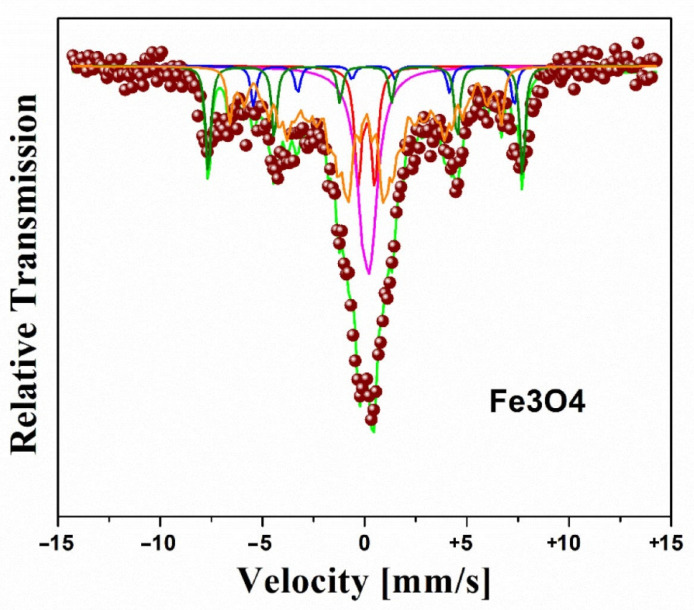
Mössbauer spectra of Fe_3_O_4_ nanoparticles at room temperature [17].

**Figure 4 molecules-27-08976-f004:**
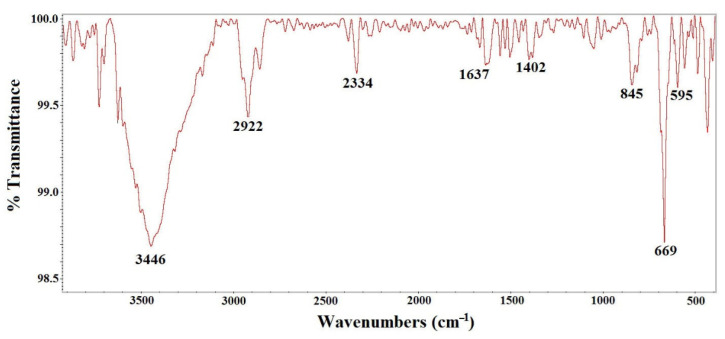
FT-IR spectrum of the Fe_3_O_4_ nanoparticles.

**Figure 5 molecules-27-08976-f005:**
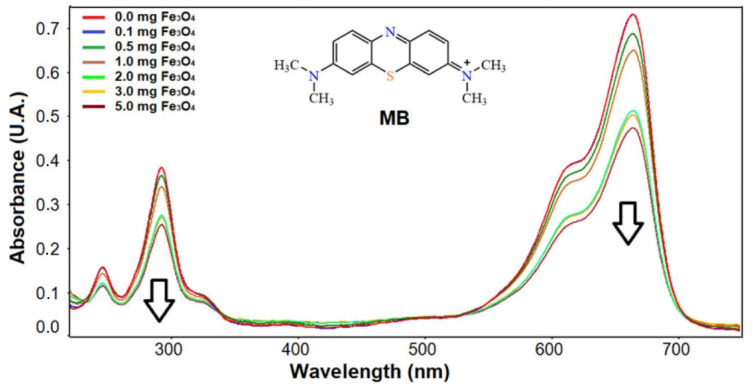
UV-Vis spectra of MB at different doses of Fe_3_O_4_ nanoparticles after 1 h.

**Figure 6 molecules-27-08976-f006:**
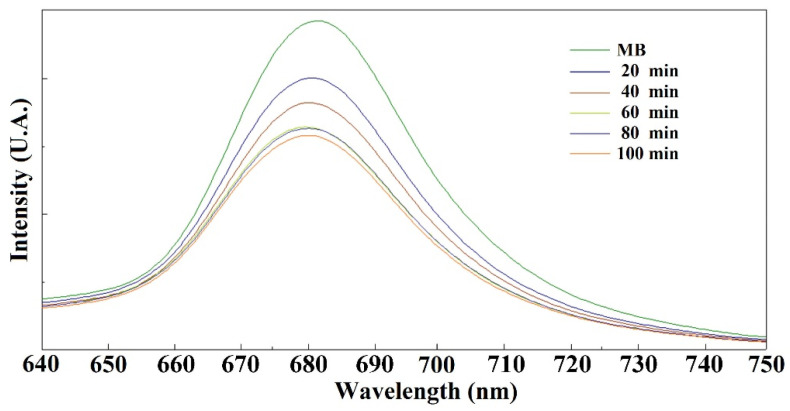
Degradation time in the presence of Fe_3_O_4_ vs. emission spectra of MB [17].

**Figure 7 molecules-27-08976-f007:**
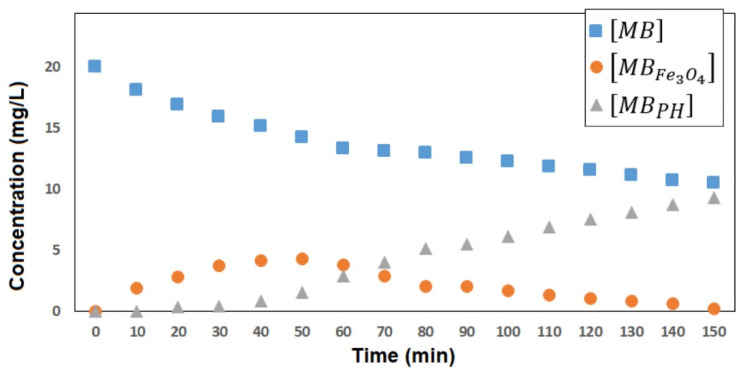
Changes of [MB], [MB_Fe3O4_], and [MB_PH_] vs. time in the photodegradation of MB at 20 mg/L catalyzed by Fe_3_O_4_ nanoparticles.

**Figure 8 molecules-27-08976-f008:**
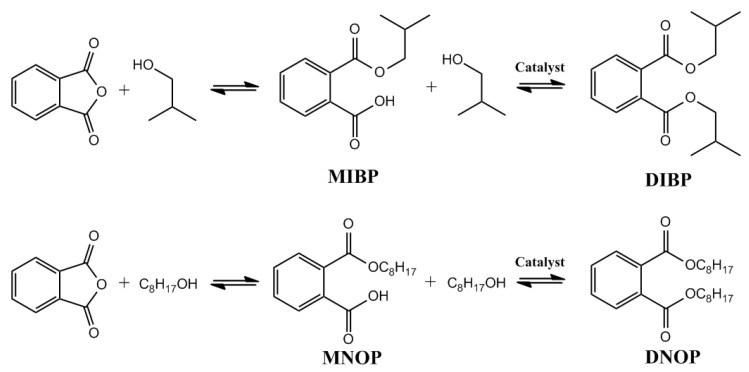
Products obtained in the esterification catalyzed by Fe_3_O_4_ nanoparticles.

**Figure 9 molecules-27-08976-f009:**
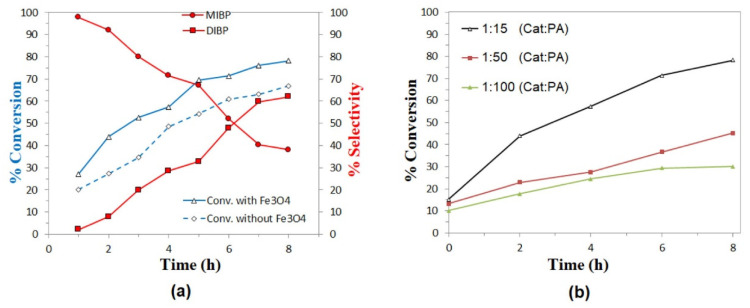
Esterification of phthalic anhydride with isobutanol: (**a**) Conversion of isobutanol and variation in the selectivity of MIBP and DIBP, as a function of time, using Fe_3_O_4_ nanoparticles as catalysts at 1:15 (catalyst:PA) molar ratio and heated at 120 °C. (**b**) Effect of catalyst:substrate ratio in conversion rates.

**Figure 10 molecules-27-08976-f010:**
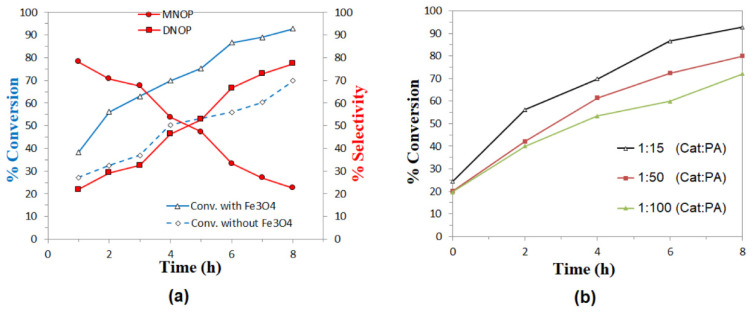
Esterification of phthalic anhydride with octanol: (**a**) Conversion of octanol and variation in the selectivity of MNOP and DNOP, as a function of time, using Fe_3_O_4_ nanoparticles as catalysts 1:15 (catalyst:PA) molar ratio and heated at 150 °C. (**b**) Effect of catalyst:substrate ratio in conversion rates.

## Data Availability

The data presented in this study were contained within the article.

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
