# Peer review of "Characterization of Fe_3_O_4_ Nanoparticles for Applications in Catalytic Activity in the Adsorption/Degradation of Methylene Blue and Esterification"

_molecules, 2022, doi:10.3390/molecules27248976_

Round 1

Reviewer 1 Report

Authors describe the synthesis of Fe3O4 particles, using them for catalytic purposes. The article lacks of more charcterization of the particles. In the keywords, Mössbauer spectroscopy is mentioned but not used.

The catalytic studies are quite poor, since there is no study of several parameters (quantity of catalyst for example is never mentioned) effect of catalyst concentration is ignored. Comaprison with similar articles with similar reactions is not mentioned.

As it is, the article cannot be published.

Author Response

Response to Reviewer 1 Comments

Point 1: Authors describe the synthesis of Fe3O4 particles, using them for catalytic purposes. The article lacks of more charcterization of the particles. In the keywords, Mössbauer spectroscopy is mentioned but not used.

The catalytic studies are quite poor, since there is no study of several parameters (quantity of catalyst for example is never mentioned) effect of catalyst concentration is ignored. Comaprison with similar articles with similar reactions is not mentioned.

As it is, the article cannot be published.

Response 1: Thank you for your comment. In section 3.2 of the Mössbauer results, the information corresponding to Mössbauer analysis was included in the manuscript. Additionally, In section 3.4 of the Catalysis studies, the information corresponding to the catalytic activity (degradation/adsorption of dye) was included. Effectively, the nanoparticles showed dye adsorption properties at the time of carrying out the study by fluorescence spectroscopy, since at the beginning a decrease in the concentration of methylene blue was observed without the formation of byproducts (confirmed by mass spectrometry). However, after several minutes of interaction of the nanoparticles with the dye, several methylene blue degradation products were identified in the solution, such as benzenesulfonic acid and thionin, molecules that have been found in other studies of the degradation of methylene blue using mass spectrometry [24], so it can be suggested that the nanoparticles presented both adsorption and catalytic degradation properties. The above clarification was included in section 3.3 (catalysis studies) of the manuscript. Besides, in section 2.1 of the Experimental Section, the information corresponding to the dose of Fe3O4 nanoparticles used in the evaluation of the catalytic activity (degradation/adsorption of dye) was included: “Finally, using a fluorescence spectrometer (FP-8500, JASCO), the degradation activity of Fe3O4 was evaluated using 2 mg of magnetite nanoparticles for every 10 mL of a cationic dye (MB) solution (20 mg/L), with the emission band at 680 nm…”

[24] Yang, C.; Dong, W.; Cui, G.; Zhao, Y.; Shi, X.; Xia, X.; Tang, B.; Wang, W. Highly efficient photocatalytic degradation of methylene blue by P2ABSA-modified TiO2 nanocomposite due to the photosensitization synergetic effect of TiO2 and P2ABSA. RSC Adv 2017, 7, 23699-23708, doi: 10.1039/C7RA02423A.

Reviewer 2 Report

In the present manuscript, authors have reported the synthesis and characterization of Fe3O4 nanoparticles. The authors have further investigated the catalytic activity of the synthesized Fe3O4 nanoparticles for the degradation of methylene blue dye and esterification reaction. The experimental data reported seems to be good and hence manuscript may be considered for publication after taking into consideration the following points.

 1) In the abstract authors claimed that synthesized Fe3O4 nanoparticles showed adsorption properties. Whereas in section 3.3 (catalysis studies) they claimed the degradation of dye. How authors justify it?

2) What dose of Fe3O4 nanoparticles was used for catalytic activity in degradation/adsorption of dye?

3) What were the criteria to choose that dose? Authors should explain the effect of changing the dose of Fe3O4 nanoparticles on degradation/adsorption of dye.

4) Suitable kinetic model can be applied to understand the rate of degradation/adsorption of dye using synthesized nanoparticles.

Author Response

Response to Reviewer 2 Comments

Point 1: In the abstract authors claimed that synthesized Fe3O4 nanoparticles showed adsorption properties. Whereas in section 3.3 (catalysis studies) they claimed the degradation of dye. How authors justify it?

Response 1: Thank you for your comment. In section 3.4 of the Catalysis studies, the information corresponding to the catalytic activity (degradation/adsorption of dye) was included. Effectively, the nanoparticles showed dye adsorption properties at the time of carrying out the study by fluorescence spectroscopy, since at the beginning a decrease in the concentration of methylene blue was observed without the formation of byproducts (confirmed by mass spectrometry). However, after several minutes of interaction of the nanoparticles with the dye, several methylene blue degradation products were identified in the solution, such as benzenesulfonic acid and thionin, molecules that have been found in other studies of the degradation of methylene blue using mass spectrometry [24], so it can be suggested that the nanoparticles presented both adsorption and catalytic degradation properties. The above clarification was included in section 3.3 (catalysis studies) of the manuscript.

[24] Yang, C.; Dong, W.; Cui, G.; Zhao, Y.; Shi, X.; Xia, X.; Tang, B.; Wang, W. Highly efficient photocatalytic degradation of methylene blue by P2ABSA-modified TiO2 nanocomposite due to the photosensitization synergetic effect of TiO2 and P2ABSA. RSC Adv 2017, 7, 23699-23708, doi: 10.1039/C7RA02423A.

Point 2: What dose of Fe3O4 nanoparticles was used for catalytic activity in degradation/adsorption of dye?

Response 2: Thank you for your comment. In section 2.1 of the Experimental Section, the information corresponding to the dose of Fe3O4 nanoparticles used in the evaluation of the catalytic activity (degradation/adsorption of dye) was included: “Finally, using a fluorescence spectrometer (FP-8500, JASCO), the degradation activity of Fe3O4 was evaluated using 2 mg of magnetite nanoparticles for every 10 mL of a cationic dye (MB) solution (20 mg/L), with the emission band at 680 nm…”

Point 3: What were the criteria to choose that dose? Authors should explain the effect of changing the dose of Fe3O4 nanoparticles on degradation/adsorption of dye.

Response 3: Thank you for your comment. The study of the degradation/adsorption of the dye was carried out at a single dose (2 mg of Fe3O4 nanoparticles per 10 mL of MB solution), so it is not possible to explain the effect of the dose change. In the present manuscript, the activity is reported with respect to the interaction time as a variable. We hope that these results can be the basis for the development of future studies that include changing the dose of nanoparticles.

Point 4: Suitable kinetic model can be applied to understand the rate of degradation/adsorption of dye using synthesized nanoparticles.

Response 4: Thank you for your comment. The objective of the study was to obtain preliminary information to design new experiments with modification of the variables. We hope that in future studies we will be able to obtain results that allow us to apply a suitable kinetic model to understand the rate of degradation/adsorption of dye using synthesized nanoparticles.

Round 2

Reviewer 1 Report

Authors corrected smoothly according to the comments.

Introduction: Some sentences might be corrected.

"The degree of structural order and size of the crystal are variable and depend on the physical and chemical properties" : Shouldn't it be the reverse ? The properties depend on order and not reverse...

"Magnetite nanoparticles exhibit a large surface area that allows them to be functionalized to make biocompatible products" : Are those particles porous to assume this sentence ?

"Another catalytic application is synthesis processes" This sentence is unclear and should be improved

Mossbauer is added but not commented as it should be. Fe3O4 is known from decades and analyses exist. Authors added only one reference for it. l. 93-98 : Those assumptions are without references.

For the catalytic experiments, how authors could quantify all the products in the reaction ?

Define HFMD

line 61: What is the dispersing agent ?

Fig.2 does not bring informations.

Fig 3: Caption has a typing mistake (tempereture)

Author Response

Response to Reviewer 1 Comments

Point 1: Introduction: Some sentences might be corrected.

"The degree of structural order and size of the crystal are variable and depend on the physical and chemical properties" : Shouldn't it be the reverse ? The properties depend on order and not reverse..

Response: The sentence was corrected in the introduction: “Physical and chemical properties depend on variables such as the degree of structural or-der and size of the crystal”.

"Magnetite nanoparticles exhibit a large surface area that allows them to be functionalized to make biocompatible products" : Are those particles porous to assume this sentence ?

Response: The sentence was corrected in the introduction: Magnetite nanoparticles exhibit a large surface area that allows them to be functionalized to make biocompatible products (porous surface)

"Another catalytic application is synthesis processes" This sentence is unclear and should be improved”

Response: The sentence was corrected in the introduction: “Within the catalytic applications are those related to chemical synthesis processes, the use of metal oxide nanoparticles as catalysts in esterification reactions has been evidenced”

Point 2: Mossbauer is added but not commented as it should be. Fe3O4 is known from decades and analyses exist. Authors added only one reference for it. l. 93-98 : Those assumptions are without references.

Response: The sentence in the introduction has been corrected and new references have been included: Mössbauer spectroscopy is a technique that can provide information on the magnetic phases and types of oxides present in the material, due to the changes generated by different elements present in the sample in the energy levels of the iron (Fe) nucleus [19-21]. Mössbauer studies have shown that magnetite (Fe3O4) spectra taken at room temperature show magnetic (ferromagnetic) behavior represented by two sextets, characteristic of the Fe+2 (octahedral sites) and Fe+3 (tetrahedral sites) states, for larger particle sizes (greater than 100 nm) [20]. However, the spectrum shows a non-magnetic (paramagnetic - superparamagnetic) behavior, represented by a doublet or singlet for particle sizes below a critical size (smaller than 20 nm) [21].

  1. Johnson, C.E.; Johnson, J.A.; Hah, H.Y.; Cole, M.; Gray, S.; Kolesnichenko, V.; Kucheryavy, P.; Goloverda, G. Mössbauer studies of stoichiometry of Fe3O4: Characterization of nanoparticles for biomedical applications. Hyperfine Interactions 2016, 237:27, 1–10, doi:10.1007/s10751-016-1277-6.
  2. Kamzin, A.S. Mössbauer Investigations of Fe and Fe3O4 Magnetic Nanoparticles for Hyperthermia Applications. Physics of the Solid State 2016, 58, 532–539, doi:10.1134/S1063783416030161.
  3. Wareppam, B.; Kuzmann, E.; Garg, V.K.; Singh, L.H. Mössbauer spectroscopic investigations on iron oxides and modified nanostructures: A review. Journal of Materials Research 2022, doi:10.1557/s43578-022-00665-4.

Point 3: For the catalytic experiments, how authors could quantify all the products in the reaction ?

Response: Thank you for your comment. The explanation of the procedure to quantify the esterification reaction products was added in section 2.1 of the manuscript, where it is indicated that the analysis of the products was carried out using gas chromatography with the help of a mass spectrometer: “The catalytic esterification reactions were carried out in a two-necked flask; phthalic anhydride (650 mg) and Fe3O4 nanoparticles (7.5 mg) were added into 2 mL of isobutanol or octanol, the mixture was heated at 120 and 150 °C respectively for several times under stirring. Analysis of the content of the reaction mixtures were made by using a Hewlett Packard 6890 gas chromatograph with a flame ionization detector. The reaction products were identified by mass spectrometry (Shimadzu-GCMS-QP2010) by Electronic Impact ionization at 70 ev.”

Point 4: Define HFMD

Response: The acronym HMFD is defined in the abstract: “hyperfine magnetic field distribution”

Point 5: What is the dispersing agent ?

Response: The dispersing agent is chloroform. Clarification left in text: “The particles were isolated by centrifugation and the precipitate appeared upon addition of dispersing agent (chloroform, 0.02 ml).”

Point 6: Fig.2 does not bring informations.

Response: We have changed figure 2 to a better resolution

Point 7: Caption has a typing mistake (tempereture)

Response: Thank you for your comment. The name of figure 3 was modified.

Reviewer 2 Report

Authors response to queries 1 & 2 is satisfactory. However, Authors have not incorporated the suggested changes in point 3 & 4 which are important to make this study meaningful. I regret to say that this manuscript cannot be accepted without addition of suggested changes in point 3 & 4.

Author Response

Response to Reviewer 2 Comments

Point 1: Authors response to queries 1 & 2 is satisfactory. However, Authors have not incorporated the suggested changes in point 3 & 4 which are important to make this study meaningful. I regret to say that this manuscript cannot be accepted without addition of suggested changes in point 3 & 4.

3) What were the criteria to choose that dose? Authors should explain the effect of changing the dose of Fe3O4 nanoparticles on degradation/adsorption of dye.

Response: To choose the dose of Fe3O4 nanoparticles on dye degradation/adsorption, it was necessary to perform UV-Vis measurements, which are presented in section 3.4 (Catalysis studies): “To carry out this study over time, it was necessary to choose the optimal dose of nanoparticles, which was achieved by varying the Fe3O4/MB ratio in different experiments by UV-Vis spectrophotometry for 1 hour of interaction. Figure 5 shows that from 2 mg Fe3O4/ 10 mL MB solution (20 mg/L), there is a significant change in the decrease in absorption corresponding to the presence of the dye, therefore the following studies were carried out using this dose”.

4) Suitable kinetic model can be applied to understand the rate of degradation/adsorption of dye using synthesized nanoparticles.

Response: A kinetic model was applied considering the adsorption/degradation processes separately in order to understand the adsorption/degradation rate of the methylene blue dye when using the synthesized Fe3O4 nanoparticles. The results and discussion can be seen in section 2.4. (Catalysis studies).

Round 3

Reviewer 1 Report

Authors corrected some points but article needs more corrections.

In introduction

"Within the catalytic applications are those related to chemical synthesis processes, the use of metal oxide nanoparticles as catalysts in esterification reactions has been evidenced" : Sentence unclear

"In applications for the removal of contaminants containing iron oxide (Fe3O4) nanoparticles, Mössbauer spectroscopy is a technique that can provide information on the magnetic phases and types of oxides present in the material, due to the changes generated by different elements present in the sample in the energy levels of the iron (Fe) nucleus" : Meaning unclear. Mössbauer spectroscopy has no influence on catalytic activity. To rephrase.

Fig.2 does not bring any information.

Fig.3 should be of better quality

fig.5 - In caption, instead of "for", after would be more relevant ?

Fig; 6 : Arrow on spectrum is not necessary

All this has to be corrected

Author Response

Response to Reviewer 1 Comments

Point 1: Introduction: Some sentences might be corrected.

"Within the catalytic applications are those related to chemical synthesis processes, the use of metal oxide nanoparticles as catalysts in esterification reactions has been evidenced" : Sentence unclear Response: Thank you for your comment. The sentence was corrected in the introduction: In many chemical synthesis processes, the use of catalysts is necessary to carry out a specific reaction, so both conversion and selectivity are important criteria to evaluate the activity of a catalyst. Among the chemical reactions that use catalysts are esterification reactions where an ester and water are obtained from a carboxylic acid and an alcohol. The use of metal oxide nanoparticles as catalysts in esterification reactions has been evidenced

"In applications for the removal of contaminants containing iron oxide (Fe3O4) nanoparticles, Mössbauer spectroscopy is a technique that can provide information on the magnetic phases and types of oxides present in the material, due to the changes generated by different elements present in the sample in the energy levels of the iron (Fe) nucleus" : Meaning unclear. Mössbauer spectroscopy has no influence on catalytic activity. To rephrase.

Response: Thank you for your comment. The sentence was corrected in the introduction: In applications for the removal of contaminants containing iron oxide (Fe3O4) nanoparticles, Mössbauer studies have shown that magnetite (Fe3O4) spectra taken at room temperature show magnetic (ferromagnetic) behavior represented by two sextets, characteristic of the Fe+2 (octahedral sites) and Fe+3 (tetrahedral sites) states, for larger particle sizes (greater than 100 nm).

Point 2: Fig.2 does not bring any information.

Response: Thank you for your comment. Unfortunately, we do not have more TEM images to add to the manuscript. The main objective of our work was to demonstrate the nanometer size of our Fe3O4 samples, which is obtained due to our production process.

Point 3: Fig.3 should be of better quality

Response: Thank you for your comment. We have already improved the figure to better resolution.

Point 4: Fig.5 - In caption, instead of "for", after would be more relevant ?

Response: Thank you for your comment. We have already changed "for" to "after".

Point 5: Fig; 6 : Arrow on spectrum is not necessary

Response: Thank you for your comment. We have already removed the arrow from the figure.

Reviewer 2 Report

Authors have incorporated the suggested changes in the revised manuscript. Manuscript may be accepted for publication after few minor revisions:

 1) Page 6, line 150, The rate at which…

2) Page 6, line 152, remove minus (-) on right hand side of eq. 3.

3) Page 7, line 156, Replace [MB]f with [MB]t on left hand side of eq. 6.

4) Page 10, lines 250-253, The sentence “The results obtained by……… wastewater decontamination.” should be revised in accordance with the kinetic model applied to adsorption/degradation results.

Author Response

Response to Reviewer 2 Comments

Point 1: Authors have incorporated the suggested changes in the revised manuscript. Manuscript may be accepted for publication after few minor revisions:

 1) Page 6, line 150, The rate at which…

2) Page 6, line 152, remove minus (-) on right hand side of eq. 3.

3) Page 7, line 156, Replace [MB]f with [MB]t on left hand side of eq. 6.

4) Page 10, lines 250-253, The sentence “The results obtained by……… wastewater decontamination.” should be revised in accordance with the kinetic model applied to adsorption/degradation results.

Response: Thank you for your comment. We have already made all the changes suggested by the referee
